# Consistency Diffusion Models for Singel-Image 3D Reconstruction with priors

## Abstract

This paper delves into the study of 3D point cloud reconstruction from a single image. Our objective is to develop the Consistency Diffusion Model, exploring synergistic 2D and 3D priors in the Bayesian framework to ensure superior consistency in the reconstruction process, a challenging yet critical requirement in this field. Specifically, we introduce a pioneering training framework under diffusion models that brings two key innovations. First, we convert 3D structural priors derived from the initial 3D point cloud as a bound term to increase evidence in the variational Bayesian framework, leveraging these robust intrinsic priors to tightly govern the diffusion training process and bolster consistency in reconstruction. Second, we extract and incorporate 2D priors from the single input image, projecting them onto the 3D point cloud to enrich the guidance for diffusion training. Our framework not only sidesteps potential model learning shifts that may arise from directly imposing additional constraints during training but also precisely transposes the 2D priors into the 3D domain. Extensive experimental evaluations reveal that our approach sets new benchmarks in both synthetic and real-world datasets. The code will be released.

## 1 Introduction

3D object reconstruction has been a long-standing challenge in computer vision, yet serving as a critical component in many real-world applications, such as robotic control (Christen et al., 2023), human-computer interaction (Liu et al., 2022b; Taheri et al., 2020), and 3D object editing (Chen et al., 2023). Once multiple views of 2D images are available, current reconstruction methods have shown superior performance. However, in extreme cases where only one single-view 2D image is provided, the limited priors often lead to significant structural ambiguity and deficiencies in the reconstructed outputs.

In recent years, a significant amount of research has focused on using traditional convolutional models for 3D reconstruction tasks from one single image (Jang & Agapito, 2021; Lin et al., 2019; Mescheder et al., 2019; Wallace & Hariharan, 2019; Yu et al., 2021). These methods typically reconstruct 3D objects in a voxelized form based on information from an image. However, such approaches often result in small-scale, low-resolution voxel representations, limiting the quality and detail of the reconstructed objects. Recently, some works (Yu et al., 2021; Henzler et al., 2021; Rematas et al., 2021; Jang & Agapito, 2021) have also utilized implicit representations and radiance fields. These methods are capable of rendering novel views with photorealistic quality but still often fail to reconstruct the possible 3D shape distribution from just one single input image. With the rising popularity of diffusion models in 2D computer vision, $PC^2$ (Melas-Kyriazi et al., 2023) is the first to directly apply the conditional diffusion model to tackle 3D point cloud reconstruction. As shown in the right top part of Fig. 1, the single image in $PC^2$ is used as the condition, which is projected on the point cloud, to train the diffusion model for predicting the Gaussian noise. We observed that, on average, only 55% of the points in $PC^2$ have initial features before training, while the initial features of the remaining points are set to zero. Thus, using only one image as a condition often results in insufficient 2D priors and weak constraints on reconstruction consistency, thereby limiting the model's performance. A subsequent variant, BDM (Xu et al., 2024), focuses on incorporating the outputs of a pre-trained model as extra priors with the outputs of $PC^2$ model in sampling time for obtaining the final reconstruction results. The model structure of BDM is shown in the right middle

part of Fig. 1. However, the results from the pre-trained network are class-level reconstruction, and BDM adopts a random combination approach to merge the outputs of the two models. Consequently, this non-specific introduction of priors still provides weak constraint on reconstruction consistency.

In this work, we propose a novel Bayesian diffusion model, termed as Consistency Diffusion Model (CDM), which leverages both 2D and 3D priors within a Bayesian framework to enhance the consistency constraint in single-image 3D point cloud reconstruction. As depicted in the bottom-right part of Fig. 1, multi-viewpoint structural priors derived from the initial point cloud are utilized as additional object-level 3D priors. One of the key contributions of this work is the introduction of a new bound term in the function derivation. This term leverages the 3D priors to continually narrow the distribution gap between the point cloud posteriors $p_\theta(x_t)$ and priors $p_\theta(x_0)$, thereby increasing the evidence lower bound (ELBO) of reconstruction probability and strengthening consistency learning during diffusion training. Specifically, the distance between the 3D priors and 3D posteriors is calculated and used as the loss value during the model's gradient descent process. This method effectively ensures that reconstruction consistency is maintained during the reverse process at any timestep.

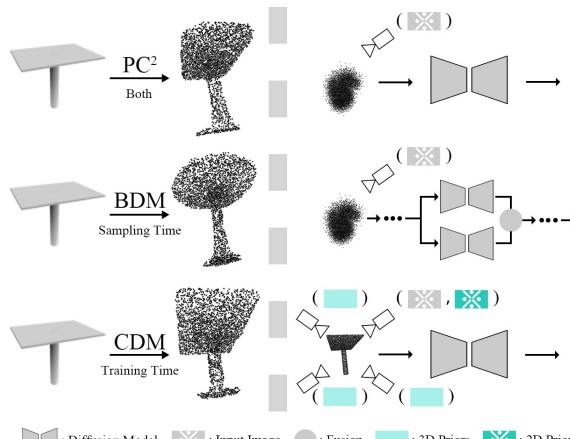

Figure 1: Illustration of reconstruction results comparison and network structures. Left: the reconstruction results of PC$^2$, BDM, and our CDM. Right: the network structures of the three approaches. BDM focuses on randomly merging the outputs of two models during the sampling phase, while our method leverages tailored 2D and 3D priors to promote consistency on the reconstruction process during training.

In terms of 2D priors, we conduct further exploration into harnessing information derived from the single image to provide more efficacious initial data throughout the training phase. Our empirical findings indicate that the incorporation of depth or contour priors extracted from the 2D image conspicuously benefits the reconstruction performance in this endeavor. Thus, while taking contour information as one prior, we also employ the DINOV2 model (Oquab et al., 2023) to process the single image, extracting the pertinent rich 2D priors. These features are subsequently mapped onto the point cloud with image features, serving to precisely regulate the training of the diffusion model.

Furthermore, we design a variety of experiments to evaluate the advantages and limitations of different approaches and strategies for this task. For instance, we comprehensively investigate critical issues such as the effectiveness of embedding information from images or text, the impact of textures and depth priors on reconstruction quality, and how 2D priors can be more effectively utilized in 3D space. Extensive experiments demonstrate that our method achieves state-of-the-art (SOTA) performance on both synthetic and real-world data.

It is worth noting that the proposed CDM relies solely on extracting 2D and 3D priors from the training data, without utilizing any auxiliary information. Additionally, empirical results demonstrate that the performance of CDM can be significantly enhanced with pretraining. The main contributions of this work are three-fold:

- Integration of 3D Priors: Leveraging intrinsic structural information from the initial point cloud as 3D priors, a new bound term is introduced in the reverse process to increase the ELBO and facilitate model convergence. This reduces uncertainty and enhances consistency in the reconstruction task.

- Exploitation of 2D Priors: Depth and contour information from the input image are utilized as 2D priors. These additional 2D priors are fused with the image features to offer more precise and effective guidance and constraints during the diffusion training.

- Empirical Validation and Superior Performance: Extensive experiments investigate the effectiveness of different types of priors and various integration strategies. Without using any auxiliary information but relying on extracting 2D and 3D priors solely from the training data, our model demonstrates high reliability and effectiveness, achieving state-of-the-art (SOTA) performance on both synthetic and real-world datasets.

## 2 RELATED WORK

### 2.1 3D SHAPE RECONSTRUCTION

In early research, 3D shapes were reconstructed by extracting multi-modal information, such as shading (Atick et al., 1996; Horn, 1970), texture (Witkin, 1981), and silhouettes (Cheung et al., 2003). With the development of neural networks, deep learning-based 3D reconstruction methods have come to dominate the field. On one hand, some works (Tatarchenko et al., 2019; Fu et al., 2021; Kato & Harada, 2019; Li et al., 2020; Fahim et al., 2021) perform 3D reconstruction task using either regression (Li et al., 2020) or retrieval (Tatarchenko et al., 2019) approaches. Other works initially leverage image geometry techniques for multi-view reconstruction (Hartley & Zisserman, 2003), then decode the extracted features using 3D convolution or sequential models to generate 3D data representations such as voxel grids. For instance, 3D-R2N2 (Choy et al., 2016a) first encodes image information into feature representations using a 2D convolution network, processes these representations with a 3D-LSTM, and finally decodes them into a voxel grid using a 3D convolution network. Pix2Vox++ (Xie et al., 2020) employs 2D convolution encoder networks and 3D convolution decoder networks, incorporating classic multi-scale feature fusion modules in its architecture. Similarly, LSM (Kar et al., 2017) utilizes a 2D network to extract image features, but it projects these 2D features into a 3D voxel grid before processing them with a 3D convolutional network.

Recently, a new research direction focusing on differentiable rendering has gained increasing popularity (e.g., NeRF (Mildenhall et al., 2021)). Most studies in this area rely on abundant multi-view data to reconstruct target scenes. However, some recent works (Chen et al., 2021; Johari et al., 2022; Kulhánek et al., 2022; Liu et al., 2022a; Henzler et al., 2021; Jang & Agapito, 2021; Rematas et al., 2021; Yu et al., 2021) have shifted focus toward learning cross-scene priors to handle the reconstruction of sparse-view scenes. Among them, works closely related to our study, such as NeRF-WCE (Henzler et al., 2021) and PixelNeRF (Yu et al., 2021), have attempted NeRF scene reconstruction from limited or single-view inputs. While these methods perform well under the few-view condition, single-view reconstruction remains a highly challenging and ill-posed problem, making it difficult for these approaches to achieve strong performance in such settings.

In our work, we adopt an entirely different approach from the aforementioned methods. We extract more initial priors to guide the training of diffusion models, enabling direct 3D point cloud reconstruction. Owing to the probabilistic nature of diffusion models, they can effectively capture the ambiguity of unseen regions while also generating high-resolution 3D point cloud shapes.

### 2.2 DIFFUSION MODEL FOR 3D RECONSTRUCTION

Image-to-3D reconstruction aims to create 3D assets from images, essentially making it a 3D generation task with 2D conditions. Recently, many works have introduced an intermediate stage that generates multi-view images before reconstructing the 3D shape. For instance, One-2-3-45 (Liu et al., 2024) leverages the 2D diffusion model Zero-1-to-3 (Liu et al., 2023b) to generate 4 posed images and trains a neural network to represent the 3D shape. One-2-345++ (Liu et al., 2023a) fine-tunes Stable-Diffusion to generate 6 posed tile images at once, improving the cross-view consistency. SyncDreamer (Liu et al., 2023c) synchronizes the intermediate states of generated multi-view images at each step of the reverse diffusion process, using a 3D-aware feature attention mechanism that correlates features across different views.

However, these approaches generally require high computational resources, and their reconstruction performance heavily depends on the quality of the generated multi-view images. This makes it challenging to precisely and effectively control the volume of the reconstructed objects. In contrast, our work eliminates the need for this intermediate multi-view generation stage. We directly generate the point cloud from a single 2D image without relying on multi-view image assistance.

### 2.3 DIFFUSION MODEL FOR 3D POINT CLOUD

Currently, 3D point cloud reconstruction remains an area in need of further exploration, with only a limited number of studies conducted. In the domain of unconditional point cloud generation, (Luo & Hu, 2021) and (Zhou et al., 2021b) design similar generation processes but use different models for diffusion training. (Lyu et al., 2021) introduced a multi-step approach by adding a refinement model after the diffusion model to further enhance the generated results, while (Vahdat et al., 2022) explored point cloud diffusion training in the latent space. However, these four works focus solely on unconditional 3D point cloud generation or completion of synthetic datasets, without addressing how to perform 3D reconstruction based on images from real-world scenes.

In the area of conditional point cloud generation, a pioneering work is PC$^2$ (Melas-Kyriazi et al., 2023), which projects encoded 2D features onto 3D point clouds as control conditions to guide the training of 3D point cloud diffusion models. The results of this work demonstrate the effectiveness of point cloud diffusion models on both real and synthetic datasets. Subsequent works have followed the structure of PC$^2$. For example, CCD-3DR (Di et al., 2023) introduces a centered diffusion probabilistic model, further improving alignment with local features. BDM (Xu et al., 2024), based on Bayesian statistical theory, employs an unconditional pre-trained diffusion model alongside the PC$^2$ diffusion model and then performs random selection to blend the outputs from both models. However, in the task of conditional 3D point cloud reconstruction, current works neither investigate 3D priors nor explore additional 2D priors, leading to weak constraints on reconstruction consistency.

Yet, rich and effective priors can often significantly enhance model performance. Therefore, we attempt to introduce rich priors from 2D diffusion models, focusing on methods for deeply exploring initial information. Our proposed framework not only integrates effective 2D priors but also excavates 3D priors from the initial point cloud, enabling effective constraints on reconstruction consistency during diffusion training.

## 3 METHOD

This section will provide a detailed explanation of our method, with Fig. 2 visually illustrating our network structure. Initially, we will briefly review the denoising diffusion model applied to point cloud data and discuss the key observations and motivations driving our approach. Subsequently, we will focus on the extraction of 3D priors from the initial point cloud and describe how we construct a bound term to align the data distributions between the posteriors and priors of the point cloud at any timestep $t$, thereby constraining the diffusion model to learn reconstruction consistency. Concluding this section, we will delve into how to further extract additional 2D priors from the image and effectively integrate them into the diffusion model for guiding the training process.

### 3.1 PRELIMINARIES OF POINT CLOUD DIFFUSION MODELS

Diffusion models serve as general-purpose generative frameworks that progressively introduce noise to a sample from a target distribution, $x_0 \sim q(x_0)$, following a series of steps determined by a variance schedule. The noise addition at each step follows a Gaussian distribution. The details of the forward and reverse process can be expressed as:

$$q(x_{1:T}|x_0) := \prod_{t=1}^{T} q(x_t|x_{t-1}), \qquad p_\theta(x_{0:T}) := p(x_T) \prod_{t=1}^{T} p_\theta(x_{t-1}|x_t). \tag{1}$$

In the context of the 3D field, a point cloud with $N$ points is treated as a $3N$ dimensional object. A diffusion model $p_\theta : \mathbb{R}^{3N} \to \mathbb{R}^{3N}$ is trained to denoise the point positions, starting from an initial Gaussian noise distribution. At each step, the network predicts the offset from the current position of each point, iteratively refining the point cloud to approximate a sample from $q(x_0)$. The network is trained by minimizing the $L_2$ loss between the predicted noise $\epsilon \in \mathbb{R}^{3N}$ and the true noise added in the time step $t$:

$$\mathcal{L} = \mathbb{E}_{\epsilon \sim \mathcal{N}(0,\mathbf{I})} \left[ \|\epsilon - p_\theta(x_t, t)\|_2^2 \right]. \tag{2}$$

PC$^2$ (Melas-Kyriazi et al., 2023) is the first work to attempt 3D point cloud reconstruction using a single 2D image as a condition within a point cloud diffusion model. This approach employs the

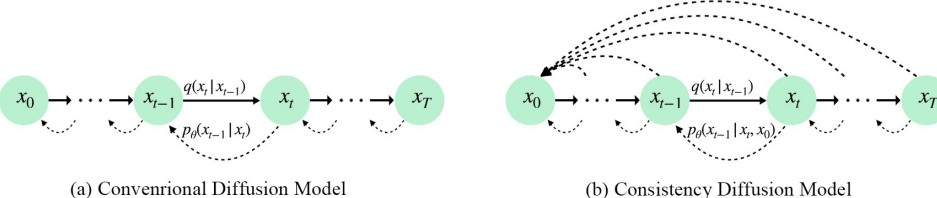

(a) Convenrional Diffusion Model                    (b) Consistency Diffusion Model

Figure 2: Model structure of conventional diffusion model and our consistency diffusion model.

camera parameters of the 2D image to rotate the noisy point cloud ($x_t$) so that it aligns with the viewpoint from which the image is captured. Subsequently, a pixel-to-point projection operation is performed on the image features, which serves as a condition to guide the diffusion training. This conditional distribution can be expressed as $q(x_0|I, V)$. The model structure of PC$^2$ is similar with subfigure (a) in Fig. 2 and the loss function can be expressed as:

$$\mathcal{L}_{\text{PC}^2} = \mathbb{E}_{\epsilon \sim \mathcal{N}(0,\mathbf{I})} \left[ \|\epsilon - p_\theta(x_t, t, I, V)\|_2^2 \right]. \tag{3}$$

Unfortunately, due to the limitation of a single viewpoint, a significant number of invisible points are assigned initial feature values of zero, which weakens the ability of the image to effectively constrain the diffusion process. The subsequent variant, BDM (Xu et al., 2024), employs a pre-trained model to generate an output, which is then randomly combined with the PC$^2$ output to obtain the final reconstructed point cloud. However, the pre-trained model only provides a class-level reconstruction, and the random combination approach still lacks a targeted constraint.

## 3.2    3D PRIORS FOR POINT CLOUD DIFFUSION MODELS

To effectively constrain diffusion training and reinforce reconstruction consistency, we propose a method that directly introduces object-level 3D priors to guide the training process. To comprehensively capture the 3D priors, we randomly apply $H$ camera rotation matrices in order to observe the initial point cloud from different viewpoints. Based on the selected rotation matrices ($R_i, T_i, i \in H$ ), the initial point cloud $x_0$ is rotated and rendered to obtain point cloud images from multiple perspectives, as shown in the upper part of Fig. 3. In this task, we hypothesize that depth information is more meaningful for 3D reconstruction (a hypothesis validated by our experimental results). Therefore, we designed a depth conversion algorithm

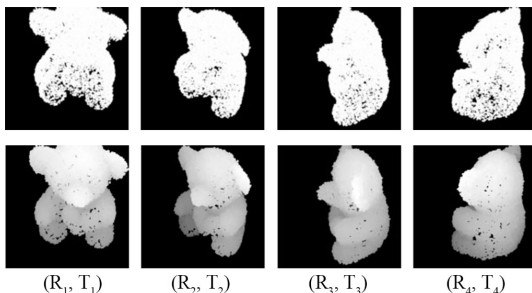

$(R_1, T_1)$  $(R_2, T_2)$  $(R_3, T_3)$  $(R_4, T_4)$

Figure 3: Illustration of rendered "teddybear" image from 4 different viewpoints. Top: rendered point cloud images. Bottom: rendered point cloud depth images.

that combines point cloud coordinates to further render point cloud depth maps, as illustrated in the lower part of Fig. 3. The depth images rendered from $x_0$ across different viewpoints serve as 3D priors to constrain the diffusion model's training.

To effectively utilize the extracted 3D priors, we have attempted different strategies. However, if we follow PC$^2$ approach, where image features are directly mapped onto the point cloud as conditions, it introduces an inconsistency in the number of conditions between the training and sampling phases, leading to model learning drift. A detailed analysis of this issue is provided in the appendix A.1.4.

To mitigate model learning drift, we strategically incorporate 3D priors as soft constraints during the training process. We formulate a bound term, named the 3D Prior Constraint, which continuously closes the data distribution between $x_t$ and $x_0$ at each timestep $t$, thereby maximizing the ELBO. For implementation, the $R_i, T_i, i \in H$ matrices defined by the 3D priors are employed to rotate the point cloud $x_t$, rendering it into $H$ point cloud depth images. Mean Square Error (MSE) is then computed between the depth images of $x_0$ and $x_t$ from the corresponding views. In our method, we retain the forward process but introduce a 3D priors constraint ($\|x_t - x_0\|^2$) to refine the reverse

Figure 4: Illustration of the detailed structure for incorporating 2D and 3D priors. The 2D priors (upper part) are concatenated with the image features and mapped onto the point cloud as conditions. The 3D priors (lower part) are transformed into depth images of the point cloud at time steps $x_0$ and $x_t$. The distances between corresponding depth images are utilized to increase ELBO.

diffusion process:

$$\tilde{p}_\theta(x_{0:T}) := p(x_T) \prod_{t=1}^{T} p_\theta(x_{t-1}|x_t) e^{-\lambda \|x_t - x_0\|^2}. \tag{4}$$

Training is performed by optimizing the variational bound on negative log-likelihood.

$$L = \mathbb{E}_q\left[-\log \tilde{p}_\theta(x_0)\right] \le \mathbb{E}_q\left[\log \frac{\tilde{p}_\theta(x_{1:T})}{q(x_{1:T}|x_0)}\right]$$

$$= \mathbb{E}_q\Big[\sum_{t=1}^{T} \underbrace{\mathcal{D}_{KL}(q(x_{t-1}|x_t,x_0)\|p_\theta(x_{t-1}|x_t))}_{L_{t-1}} + \lambda \sum_{t=1}^{T} \underbrace{\|x_t - x_0\|^2}_{L_{3D\ \text{Priors Constraint}}}$$

$$+ \underbrace{\mathcal{D}_{KL}(q(x_T|x_{t_0})\|p(x_T))}_{L_T} - \underbrace{\log p_\theta(x_0|x_T)}_{L_0}\Big] \tag{5}$$

It can be derived from the formula that adding this term increases the ELBO, thereby facilitating the convergence of the diffusion model. In this point cloud reconstruction task, enhanced model convergence implies that reconstruction consistency has been effectively improved while suppressing generation uncertainty. The results in Fig. 5 intuitively support this conclusion. Due to the inherently disordered and sparse nature of 3D point cloud (Guo et al., 2020), the bound term $\|x_t - x_0\|^2$ is intractable. Consequently, the 3D priors are converted to 2D depth images at timestep $t$ to measure the distance between $x_0$ and $x_t$. $v_i$ represents the camera parameter for viewpoint $i$. The objective after simplification is:

$$L(\theta) := \mathbb{E}_{\epsilon \sim \mathcal{N}(0,\mathbf{I})}\Big[ \underbrace{\|\epsilon - p_\theta(x_t, t, I, V)\|_2^2}_{\text{Diffusion Loss}} + \sum_{i=1}^{H} \underbrace{\|\text{proj}(x_t, v_i) - \text{proj}(x_0, v_i)\|_2^2}_{\text{Priors Constraint}}\Big] \tag{6}$$

The bound term computed between the point cloud depth maps at time step $0$ and $t$ from all $H$ viewpoints are summed up. This bound term is then incorporated into the backpropagation process as a regularization term to constrain the model training and reinforce the reconstruction consistency.

### 3.3 2D PRIORS FOR POINT CLOUD DIFFUSION MODELS

For single-image 3D point cloud reconstruction tasks, the image serves as the sole condition to guide the diffusion training, meaning that the information contained in the image directly influences the model's final performance. In various 2D vision tasks, the integration of strong priors has significantly improved model performance. Therefore, we also considered how to leverage existing models to further process the image, incorporating additional 2D priors to guide and constrain the training of the 3D point cloud diffusion model.

In terms of 2D global priors, we attempt to use OpenCLIP (Cherti et al., 2023) to process the 2D images, extracting both text and image embeddings. These embeddings are then iteratively integrated into the model using a cross-attention structure similar to that in ControlNet (Zhang et al., 2023). However, the experimental results indicate that this approach does not yield any improvements. For 2D local priors, we utilize Zero123 (Liu et al., 2023b) to generate multi-view images from the single 2D image. Unfortunately, due to the arbitrary camera angles of the 2D images, accurately estimating the camera $(R, T)$ matrix of the generated images proved challenging. Consequently, the multi-view images can not be accurately aligned with the point cloud in this task. Further analysis and results are provided in the section of Experiment 4.2 and Appendix A.1.3.

Based on multiple attempts and experiments, we analyze that the 2D priors can only be extracted from the initial 2D image ($I$) and that the additional 2D priors can be integrated into the model training by overlaying them with the initial image features. Currently, the image information is primarily extracted using a pre-trained Vision Transformer (ViT) network (Dosovitskiy, 2020), which captures features that mainly reflect planar texture characteristics. From our findings during the process of incorporating 3D priors, we believe that depth information can provide more reliable guidance for 3D reconstruction. Therefore, we delve into the depth information contained in the initial 2D image. Utilizing the DINOV2 (Oquab et al., 2023) model, we perform depth or contour estimation on $I$, and then we overlay this information as an additional 2D priors with the features outputted from the ViT, using concatenation for the integration of the 2D priors.

$$\text{2D Priors} = F_I \oplus F_{I^*}, \tag{7}$$

where $F_{I^*}$ represents the outputs of DINOV2. The top part of Fig. 4 illustrates the process of 2D priors incorporation, which facilitates the subsequent pixel-to-point mapping operation to assist in noise prediction. This approach effectively introduces additional 2D priors as a condition to guide the diffusion training. In the Tab. 4, the results clearly show that depth and contour information provide valuable guidance for reconstructing the point cloud, leading to improved performance.

## 4 EXPERIMENTS

**Dataset.** To evaluate the effectiveness of our proposed method, we conduct experiments on two distinct datasets: the synthetic dataset ShapeNet (Choy et al., 2016b; Chang et al., 2015) and the real-world dataset Co3D (Reizenstein et al., 2021). ShapeNet dataset is a comprehensive collection of 3D computer-aided design models, covering 3,315 categories derived from the WordNet (Miller, 1994) database. In contrast, the Co3D dataset presents a challenging benchmark, as it consists of multi-view images of real-world objects from common object categories. We compare our results with PC$^2$ and BDM. Due to the high computational cost of using a class-level pre-trained model, BDM conducts experiments on only five categories in the ShapeNet dataset. Additionally, BDM does not perform experiments on the Co3D dataset and does not provide checkpoints for the relevant pre-trained models. Our experimental setup is consistent with that of prior works in terms of image rendering, camera matrices, and train-test splits, thereby ensuring a fair and comparable evaluation of the methodologies employed.

**Metrics.** The effectiveness of the reconstruction process is evaluated using two widely accepted performance metrics: Chamfer Distance (CD) and F-Score@0.01 (F1). The Chamfer Distance quantifies the discrepancy between two point sets by calculating the shortest distance from each predicted point to its nearest point in the ground truth. To mitigate CD's sensitivity to outliers, we also report the F-Score at a threshold of 0.01. In this metric, a reconstructed point is deemed accurately predicted if its nearest distance to any point in the ground truth point cloud falls within the specified threshold, providing a measure of precision in the reconstruction process.

**Implementation Details.** The aim is to maintain consistency across all settings in accordance with previous works. All images and rendering resolutions are set to $224 \times 224$ pixels. For the ShapeNet dataset, a total of 4,096 points are sampled for each 3D object, while for the Co3D dataset, 16,384 points are employed. Our method is implemented based on PC$^2$ using PyTorch. The PyTorch3D library is used to render the 3D prior images and handle rasterization during the projection conditioning phase. In the BDM experiments, all settings are kept at their original values, and PVD (Zhou et al., 2021a) is still utilized as the pre-trained model, with checkpoints sourced from the BDM code repository. During the 3D prior point cloud projection, we used a point size of 0.04 and projection images of $224 \times 224$ pixels. All experiments are conducted on a single GeForce RTX-4090 GPU.

## 4.1 QUANTITATIVE RESULTS

**ShapeNet.** In Tab. 1, we present the results for the 13 classes in the widely used ShapeNet dataset, compared with $PC^2$. The results show a consistent improvement over those obtained with $PC^2$ on the ShapeNet dataset. This is particularly evident in the F1 score, indicating that our method demonstrates superior reconstruction capabilities across most categories. While the Chamfer Distance is generally lower for our method in several cases, the differences are either minor or slightly favor $PC^2$.

As for Tab. 2, we compare our CMD with $PC^2$ and BDM. The BDM framework enables a reconstruction model and a pre-trained model to be sampled together during inference, incorporating random selection at some intermediate steps for fusing the two resulting point clouds. Due to the high computational cost of using a class-level pre-trained model, BDM only conducts experiments on five categories. In this ex-

Table 1: Performance comparison on ShapeNet.

|  | $PC^2$ | | CDM | |
|---|---|---|---|---|
|  | CD↓ | F1↑ | CD↓ | F1↑ |
| airplane | 65.97 | 0.655 | **59.72** | **0.660** |
| bench | **46.16** | 0.658 | 48.14 | **0.671** |
| cabinet | 54.87 | 0.454 | **53.54** | **0.464** |
| car | 64.36 | 0.547 | **63.92** | **0.558** |
| chair | 65.57 | 0.464 | **62.93** | **0.476** |
| display | 79.64 | 0.537 | **77.48** | **0.549** |
| lamp | 132.44 | 0.437 | **125.41** | **0.448** |
| loudspeaker | **83.79** | 0.392 | 84.74 | **0.393** |
| rifle | 29.37 | 0.776 | **27.20** | **0.791** |
| sofa | 47.54 | 0.472 | **45.33** | **0.492** |
| table | 73.87 | 0.527 | **67.95** | **0.547** |
| telephone | **48.33** | 0.671 | 49.61 | **0.674** |
| watercraft | 48.26 | 0.574 | **46.83** | **0.586** |
| *Average* | 64.63 | 0.551 | **62.52** | **0.562** |

periment, our model also attempted to incorporate the priors from BDM's pre-trained model. The results indicate that our model achieves the best performance without using the pre-trained model priors, and incorporating the pre-trained model can further improve the reconstruction results.

Table 2: Performance comparison on ShapeNet. +BDM means using BDM during sampling.

|  | airplane | | car | | chair | | sofa | | table | | *Average* | |
|---|---|---|---|---|---|---|---|---|---|---|---|---|
|  | CD↓ | F1↑ | CD↓ | F1↑ | CD↓ | F1↑ | CD↓ | F1↑ | CD↓ | F1↑ | CD↓ | F1↑ |
| $PC^2$ | 65.97 | 0.655 | 64.36 | 0.547 | 65.57 | 0.464 | 47.54 | 0.472 | 73.87 | 0.527 | 63.46 | 0.533 |
| $PC^2$+BDM | 59.04 | 0.660 | 65.85 | 0.559 | 64.21 | 0.485 | 44.12 | 0.504 | 68.35 | 0.551 | 60.31 | 0.552 |
| CDM | 59.72 | 0.660 | 63.92 | 0.558 | 62.93 | 0.476 | 45.33 | 0.492 | 67.95 | 0.547 | 59.97 | 0.547 |
| CDM+BDM | **57.35** | **0.665** | **60.44** | **0.569** | **58.54** | **0.497** | **42.81** | **0.512** | **64.17** | **0.568** | **56.66** | **0.562** |

**Co3D.** In real-world scenarios, the 3D reconstruction performance of our method is illustrated in Tab. 3. The objects exhibit greater detail in their shapes and more intricate geometric configurations. We conducted experiments on the challenging Co3D dataset. Since BDM does not provide the relevant pre-trained models and has not tested on this dataset, we only compared our method with $PC^2$ in this section, focusing on three categories. This outcome demonstrates the effectiveness of our method

Table 3: Performance comparison on Co3D.

|  | $PC^2$ | | CDM | |
|---|---|---|---|---|
|  | CD↓ | F1↑ | CD↓ | F1↑ |
| hydrant | 92.32 | 0.485 | **90.11** | **0.507** |
| teddybear | 116.95 | 0.428 | **102.79** | **0.461** |
| toytruck | 153.84 | 0.421 | **125.40** | **0.474** |
| *Average* | 121.04 | 0.445 | **106.10** | **0.481** |

in addressing the challenges posed by real-world objects. Our proposed method shows superior performance across all object categories in both Chamfer Distance and F1 Score, indicating better geometric accuracy and precision in the 3D reconstructed models using just a single image. In Appendix A.1.1, we provide more visual comparison results in Fig. 6 to intuitively demonstrate our model's superior performance in reconstruction consistency.

**Visualization.** In this experiment, we present visual comparison results on the ShapeNet dataset. We compare our method with PC2 and BDM across three categories of reconstruction. The first column of Fig. 5 displays the input images, and we compare the reconstruction results from two different viewpoints. Intuitively, the results from PC2 exhibit ambiguities due to a lack of priors. For instance, in the first row, the sofa has two backs, and in the third row, the table appears to have two layers. In the case of BDM, the reconstruction results are significantly influenced by the non-

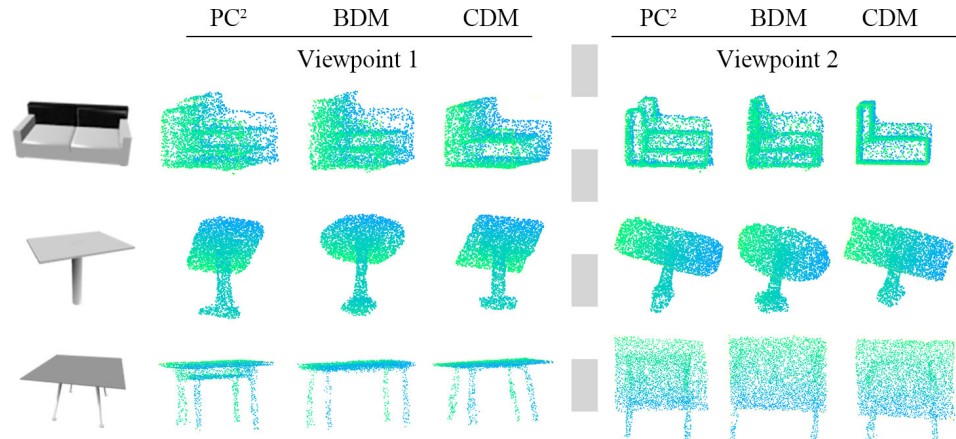

Figure 5: Visual comparison on the ShapeNet dataset. The first column displays the input image. We compare the reconstructed point clouds from two different viewpoints. Intuitively, PC² produces ambiguous results due to weak constraints, and BDM, which introduces class-level priors, fails to effectively control reconstruction consistency.

specific object information introduced by class-level priors. For example, in the second row, a round tabletop is reconstructed, while in the third row, the table legs are spaced too far apart. In contrast, the object-level constraints of our method lead to higher consistency between the reconstruction results and the input images.

## 4.2 ABLATION STUDY

To validate the effectiveness and rationality of our method, we conducted a series of ablation studies to investigate the influence of both 2D and 3D priors. For simplicity, the results of these ablation experiments are evaluated using the Co3D "Teddybear" category.

Table 4: Ablation study of leveraging 2D and 3D priors.

|  | teddybear | | toytruck | | hydrant | | Average | |
|---|---|---|---|---|---|---|---|---|
|  | CD $\downarrow$ | F1 $\uparrow$ | CD $\downarrow$ | F1 $\uparrow$ | CD $\downarrow$ | F1 $\uparrow$ | CD $\downarrow$ | F1 $\uparrow$ |
| Baseline (w/o prior) | 116.95 | 0.428 | 153.84 | 0.421 | 92.32 | 0.485 | 121.04 | 0.445 |
| 2D Prior | 107.09 | 0.448 | 148.76 | 0.466 | 90.67 | 0.527 | 115.51 | 0.480 |
| 3D Prior | 106.35 | 0.446 | 135.38 | 0.463 | **90.33** | 0.534 | 110.69 | 0.481 |
| 2D+3D Prior | **102.79** | **0.461** | **125.40** | **0.474** | 93.57 | **0.536** | **107.25** | **0.490** |

**Leverage 2D and 3D Priors.** We first verify the effectiveness of 2D and 3D priors. In Tab. 4, utilizing either 2D or 3D priors leads to improved performance across the three categories in the Co3D dataset. This highlights the complementary strengths of 2D and 3D information in enhancing object reconstruction quality. Furthermore, the combined priors configuration (2D + 3D) consistently performs well across all individual categories, as evidenced by its superior performance in both Chamfer Distance (CD) and F1 metrics compared to other prior configurations.

**3D Prior Frames and Point Size.** Tab. 5 investigates the impact of the number of prior frames and point size on model performance. If the point size is insufficient, it can result in a sparse point cloud rendering, which may lead to inaccurate distance calculations during projection rendering between point cloud $x_0$ and $x_t$. The number of frames is also a crucial factor in enhancing model efficacy. Given the inherent disorder of the point cloud, we use the rendering process to describe the shape in question. Increas-

Table 5: The impact of the number of camera rotations matrix ($H$) and rendered point size.

| Settings | F1 $\uparrow$ |
|---|---|
| 4 Frames + 0.04 point size | 0.452 |
| 10 Frames + 0.0075 point size | 0.451 |
| 10 Frames + 0.04 point size | 0.461 |

ing the number of point cloud frame renderings allows for a more comprehensive description of the point cloud shape from a broader range of viewpoints.

**Image and Text Embedding for Global Features.** Tab. 6 presents an ablation study on the performance of various global feature conditioning methods. We explore two approaches: concatenation ($\oplus$) and cross-attention ($\otimes$) of different features. The results indicate that both concatenation and cross-attention tend to degrade model performance. These findings suggest that local features may contribute more effectively to model performance in this context. In our method, the image encoder extracts the local features from the image, while the diffusion backbone extracts both local and global features from the point cloud.

Table 6: Comparison of different types of features and utilization strategies. $\oplus$ means concatenation and $\otimes$ means cross attention.

| Diffusion Model Conditions | F1 $\uparrow$ |
|---|---|
| Image feature | 0.428 |
| Image feature $\oplus$ OpenCLIP (text) | 0.230 |
| Image feature $\oplus$ OpenCLIP (image) | 0.320 |
| Image feature $\oplus$ OpenCLIP (depth image) | 0.421 |
| Image feature $\otimes$ OpenCLIP (image) | 0.423 |
| Image feature $\otimes$ OpenCLIP (text) | 0.379 |

Table 7: Comparison of 2D priors utilization strategies during the training.

| Conditions | F1 $\uparrow$ |
|---|---|
| 1 GT image | 0.428 |
| 1 GT image and 3 ControlNet images | 0.423 |
| 1 GT image and 1-3 ControlNet | 0.416 |
| 4 Gray ControlNet images | 0.425 |
| 4 GT images | 0.420 |
| 1 ControlNet image | 0.413 |
| 1 ControlNet Gray image | 0.425 |

**More Conditions in Training.** As demonstrated in Tab. 7, utilizing varying numbers of input images as conditions yields distinct outcomes. The results indicate that directly projecting the additional 2D priors (results generated by ControlNet (Zhang et al., 2023)) onto the point cloud leads to model learning drift, as illustrated by the results in rows 2, 3, and 4. Even when using only GT images for projection, there is no improvement in performance, as shown in row 5. If the input single image is replaced with one generated by ControlNet, the inaccuracies in the generated image features result in a decline in performance. A detailed description of the experiments using images generated by ControlNet is provided in the Appendix A.1.4.

**Rendering Methods of Point Cloud.** In Tab. 8, we compare the effects of different 2D priors (depth and contour) on the reconstruction task. The experimental results indicate that using contour as a 2D prior yields better performance on the ShapeNet dataset, while depth proves to be a more effective 2D prior on the Co3D dataset. We attribute this difference to the fundamental characteristics of the two datasets. ShapeNet is an artificially synthesized dataset, which can introduce biases in depth information extraction, making the more accurate contour information more beneficial for performance. In contrast, Co3D comprises images from real-world scenes, where accurate depth information is more advantageous for reconstruction.

Table 8: Comparison of point cloud rendering methods. Airplane and chair are the ShapeNet category and Teddybear is in the Co3D category.

| | Contour | | Depth | |
|---|---|---|---|---|
| | CD $\downarrow$ | F1 $\uparrow$ | CD $\downarrow$ | F1 $\uparrow$ |
| ShapeNet | | | | |
| airplane | **59.72** | **0.660** | 62.37 | 0.655 |
| chair | **62.93** | **0.476** | 63.39 | 0.474 |
| Co3D | | | | |
| teddybear | 106.48 | 0.451 | **102.79** | **0.461** |
| *Average* | 76.38 | **0.529** | **76.18** | 0.530 |

## 5 CONCLUSION

This work proposes a Consistency Diffusion Model designed to enhance the model's focus on reconstruction consistency. By extracting the inherent structural information from point cloud data, we introduce object-level 3D priors to constrain the model learning. Specifically, we propose a new bound term that leverages these 3D priors to increase the ELBO, reducing the uncertainty of the diffusion model, and reinforcing consistency. Additionally, we extract depth and contour information from the input image as additional 2D priors, effectively guiding and constraining the training process. We conducted extensive comparative experiments to evaluate the effectiveness of different priors and incorporation strategies. The experimental results consistently show that our method achieves SOTA performance in both synthetic and real-world scenarios. For future work, we plan to integrate the reconstructed point cloud with textual descriptions for point cloud editing.

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

# A  APPENDIX

## A.1  FUNCTION DERIVATION

Below is a derivation of Eq. 5, which presents the reduced variance variational bound for diffusion models in the context of our reverse process.

$$
\begin{aligned}
L &= \mathbb{E}_q \left[ -\log \tilde{p}_\theta(x_0) \right] \\
&\leq \mathbb{E}_q \left[ \log \frac{\tilde{p}_\theta(x_{1:T})}{q(x_{1:T}|x_0)} \right] \\
&= \mathbb{E}_q \left[ \sum_{t=1}^{T} \log \tilde{p}_\theta(x_{t-1}|x_t, x_0) + \log p_\theta(x_T) - \sum_{t=1}^{T} \log q(x_t|x_{t+1}) \right] \\
&= \mathbb{E}_q \left[ \sum_{t=1}^{T} \log p_\theta(x_{t-1}|x_t, x_0) + \lambda \sum_{t=1}^{T} \|x_t - x_0\|^2 + \log p_\theta(x_T) - \sum_{t=1}^{T} \log q(x_t|x_{t+1}) \right] \\
&= \mathbb{E}_q \Big[ \sum_{t=1}^{T} \underbrace{\mathcal{D}_{KL}(q(x_{t-1}|x_t, x_0)\|p_\theta(x_{t-1}|x_t))}_{L_{t-1}} + \lambda \sum_{t=1}^{T} \underbrace{\|x_t - x_0\|^2}_{L_{\text{3D Priors Constraint}}} \\
&\quad + \underbrace{\mathcal{D}_{KL}(q(x_T|x_{t_0})\|p(x_T))}_{L_T} - \underbrace{\log p_\theta(x_0|x_T)}_{L_0} \Big]
\end{aligned}
\tag{8}
$$

### A.1.1  VISUAL COMPARISON ON CO3D DATASET

Fig. 6 presents additional visual results. We compare our method with $PC^2$ on the Co3D dataset. The first column on the left displays the input images. By comparing from two different viewpoints, it is intuitively evident that $PC^2$'s reconstruction results exhibit significant ambiguity and missing parts in areas that are not visible from the viewpoint. In contrast, our method maintains strong consistency with the input images.

### A.1.2  GLOBAL PRIORS KNOWLEDGE EMBEDDING

In this work, we aim to extract global information from the 2D image. By inputting a single 2D image into OpenCLIP (Cherti et al., 2023), we obtain both text and image embeddings. We then apply a multi-scale cross-attention structure, similar to ControlNet structure, to iteratively integrate the priors from OpenCLIP into the network. Tab. 6 presents the results of embedding these global priors. From the experimental results, we observe that embedding either type of prior (text or image embedding) individually or jointly does not enhance performance and may even lead to a decline. We analyze this outcome and conclude that, for the 3D point cloud reconstruction task, global information provides only a rough understanding of the object, while detailed features are essential for effective reconstruction. Consequently, we shift our focus to exploring how to incorporate more detailed priors into the reconstruction process.

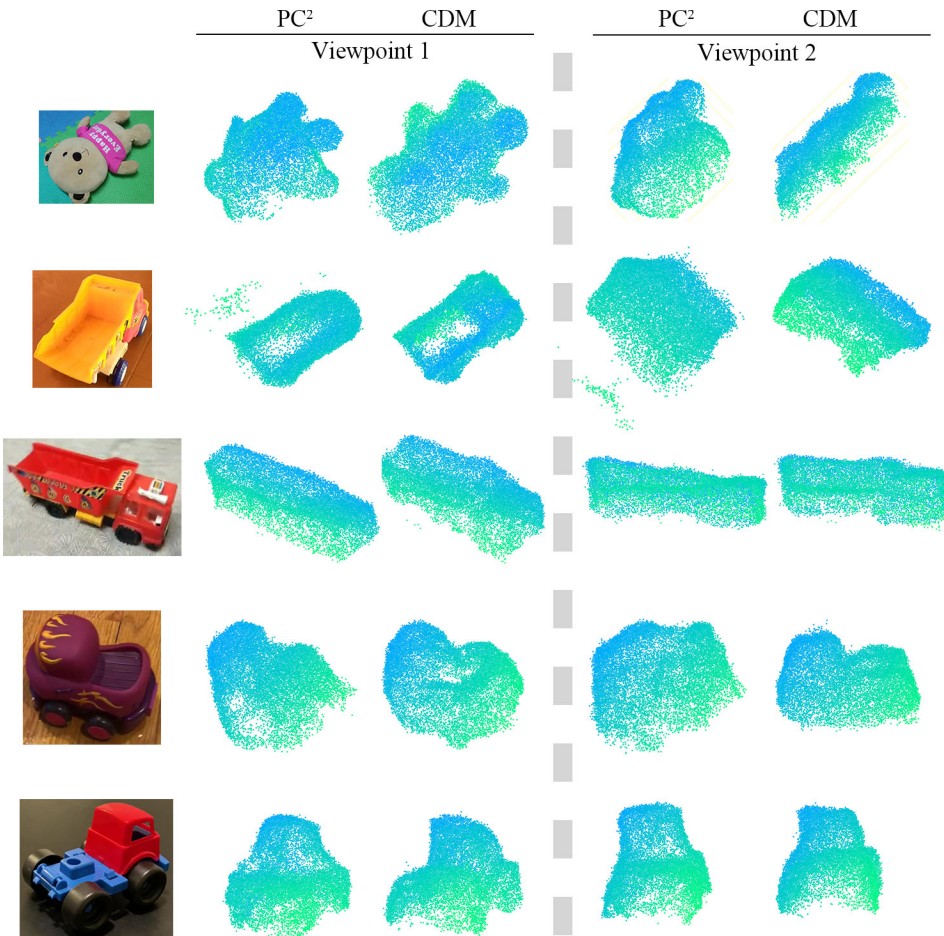

Figure 6: Visual comparison on the Co3D dataset. The first column displays the input image. We compare the reconstructed point clouds from two different viewpoints. Intuitively, $PC^2$ produces ambiguous results due to weak constraints.

### A.1.3    LOCAL PRIORS KNOWLEDGE EMBEDDING

On the local feature level, we experiment with using Zero123++ (Shi et al., 2023) to generate images of the target object from various angles based on a single 2D image. The aim is to project features from these multi-view images onto the point cloud after rotating the cloud, thereby increasing the number of points with initial features. However, during our experiments, we find that due to the arbitrary nature of the 2D image's camera parameters and the uncalibrated position of the image relative to the target object, the camera parameters of the images generated by Zero123++ are often difficult to estimate accurately. This make it challenging to rotate the point cloud to match the input image correctly, preventing effective pixel-to-point feature mapping. As shown in Fig. 7, when a single image is input, the zero123++ method can only generate reconstructed images from a fixed viewpoint, which exhibit significant deformation. Consequently, these generated images not only cannot be aligned with the point cloud using the camera rotation matrices, but they also introduce a considerable amount of erroneous information.

### A.1.4    DIRECTLY INTRODUCE 2D PRIORS

Based on our experiments with both global and local priors, we conclude that the key to effectively incorporating 2D priors is to stack these priors directly onto the single input image. Therefore, we straightforwardly follow the training approach of $PC^2$, mapping the depth map features from different angles to the point cloud through pixel-to-point projection. To ensure consistency with the original 2D image used as a condition, we fine-tuned ControlNet (Zhang et al., 2023) using multi-

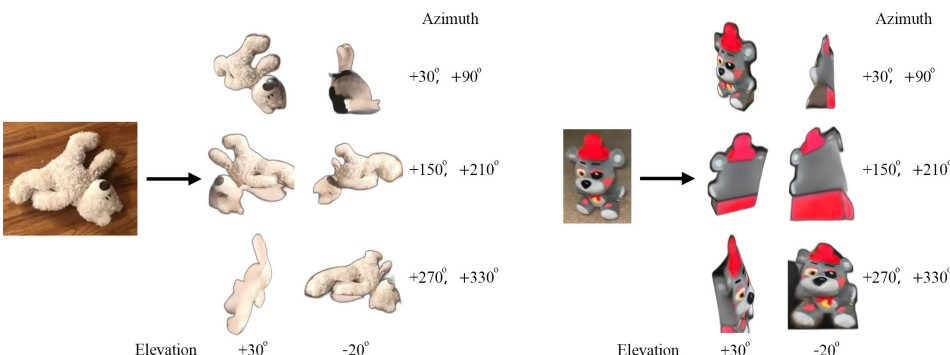

Figure 7: Illustration of the generation results of the zero123++ on the "teddybear" category in the Co3D dataset. zero123++ generates images of the target object from six fixed viewpoints. However, due to the arbitrary positioning of the target object, the generated images frequently contain ambiguities, and the object's structure appears errors.

view point cloud images. This fine-tuning enables ControlNet to generate corresponding 2D texture images from the point cloud images, as illustrated in Fig. 8. Subsequently, we used the generated 2D images to assign features to the initial point cloud $x_0$. As a result, on average, 97% of the points in $x_0$ now have initial features, significantly addressing the issue of many points having zero initial features due to occlusions from a single viewpoint. This enhancement provides a stronger constraint for reconstruction. Tab. 7 presents a comparison of the reconstruction results using this approach.

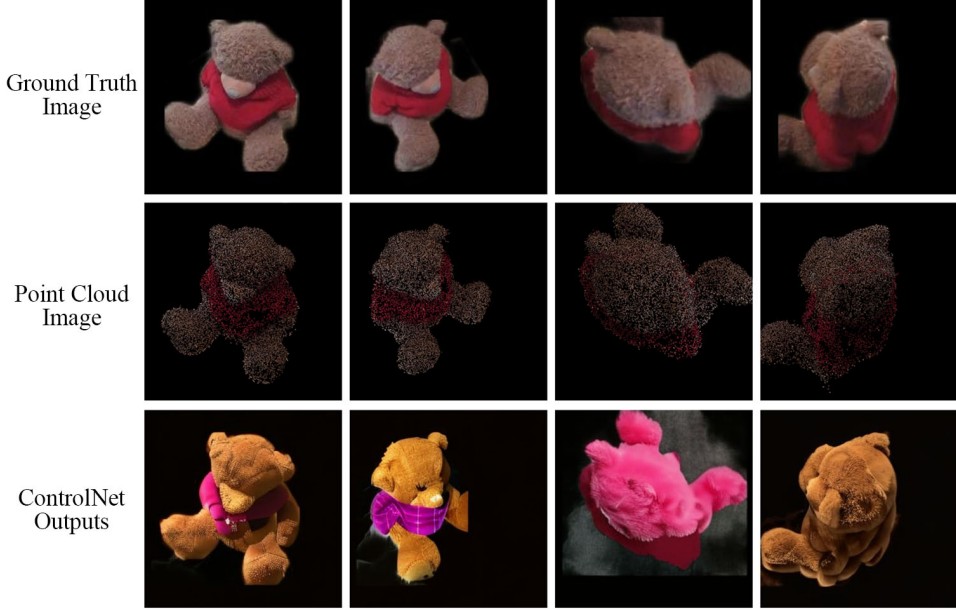

Figure 8: Illustration of ControlNet outputs after fine-tuning. The first row shows the ground truth (GT) input images, and the second row displays the rendered point cloud images from the same viewpoint. We pair the images from the first and second rows that correspond to the same viewpoint for fine-tuning ControlNet. The third row contains the outputs after fine-tuning ControlNet. It is evident that this fine-tuning approach ensures that the shapes of the output images are completely consistent with the GT images.

Through our comparison results, we observed that introducing more conditions—thus increasing the proportion of point clouds with initial features—led to a decline in model performance during sampling. This unexpected outcome drew our attention and prompted further investigation. We believe this issue arises from the mismatch in the number of conditions between the training and sampling phases, resulting in a deviation in the model's learning process. To the best of our knowledge,

no prior work has proposed or discussed the impact of inconsistent numbers of conditions during training and sampling. We refer to this resulting issue as "model learning drift." This phenomenon occurs because, during training, the model relies on multiple conditions to guide its learning effectively. However, during sampling, we lack access to the initial point cloud and cannot generate additional images through ControlNet as conditions by rotating the point cloud. Consequently, only one image is available as a condition during sampling. The absence of other control conditions during this phase contributes to the observed learning drift.

