# OpenReview forum: "Consistency Diffusion Models for Singel-Image 3D Reconstruction with priors"
_ICLR.cc/2025/Conference — ICLR 2025 Conference Withdrawn Submission_

### Official Review · Reviewer_fwgZ · 2024-10-22

**Soundness:** 3
**Presentation:** 2
**Contribution:** 2
**Rating:** 5
**Confidence:** 3

**Summary:**

This paper aims to reconstruct point cloud from a single image. First, they convert 3D structural priors derived from the initial 3D point cloud as a bound term to increase evidence in the variational Bayesian framework. Second, they extract and incorporate 2D priors from the single input image, projecting them onto the 3D point cloud to enrich the guidance for diffusion training. The results show SOTA performance.

**Strengths:**

1.This paper integrates 3D and 2D priors to the reconstruction task. The results achieve SOTA performance in ShapeNet and Co3D dataset.

**Weaknesses:**

1.The paper introduces 3D priors constraint to refine the reverse process. Does this part increase the training time of the model? It is better to give a comparison of the training time of the model with and without the 3D prior.

2.Visual comparison is not sufficient. Only three samples were selected in the qualitative experiment of the Shapenet dataset (Figure 5), and the differences are not quite visible in all the samples except for the middle sample where CDM showed advantages. Besides, the advantage of CDM over PC2 is not apparent from the Visual comparison on the Co3D dataset in Figure 6.

3.The depth map rendered from the point cloud is a random sample of the 3D geometry, and a lot of information is lost in the sampling process. It is better to directly adopt 3D geometric representation such as SDF as 3D prior.

4.The paper lacks sufficient research and comparison on relevant methods. There are a large number of methods that can reconstruct point clouds from a single image with good results. For example, TriplaneGaussian[1] can generate multi-view rendering results in addition to point clouds; Michelangelo[2] can generate point clouds corresponding to text and images; and CLAY[3] trained a large model for point cloud generation. The ability of these methods to generate point clouds from images is not limited to a few single categories, and they have good generalization ability. These methods should be discussed and compared in the paper.

[1] Zou Z X, Yu Z, Guo Y C, et al. Triplane meets gaussian splatting: Fast and generalizable single-view 3d reconstruction with transformers[C]//Proceedings of the IEEE/CVF Conference on Computer Vision and Pattern Recognition. 2024: 10324-10335.

[2] Zhao Z, Liu W, Chen X, et al. Michelangelo: Conditional 3d shape generation based on shape-image-text aligned latent representation[J]. Advances in Neural Information Processing Systems, 2024, 36.

[3] Zhang L, Wang Z, Zhang Q, et al. CLAY: A Controllable Large-scale Generative Model for Creating High-quality 3D Assets[J]. ACM Transactions on Graphics (TOG), 2024, 43(4): 1-20.

**Questions:**

1.How is the camera matrix selected? If only random sampling is used, the images rendered from adjacent views will be very similar.

2.Please refer to the questions and suggestions in the Weaknesses part.

---

### Official Review · Reviewer_Hjmg · 2024-11-03

**Soundness:** 2
**Presentation:** 3
**Contribution:** 2
**Rating:** 3
**Confidence:** 5

**Summary:**

The paper proposes the Consistency Diffusion Model (CDM) for 3D point cloud reconstruction from a single image, integrating both 2D and 3D priors to enhance consistency in the Bayesian framework. By converting 3D structural priors into a bound term, the approach aims to increase evidence in the variational Bayesian framework, and 2D priors from the input image are projected onto the 3D point cloud for additional guidance. The authors claim this framework offers a consistency advantage over existing methods and improves performance on synthetic and real-world datasets.

**Strengths:**

The paper does present an effort to enhance consistency in 3D shape reconstruction by combining 2D and 3D priors within a Bayesian framework. While the theoretical foundation is weak, the authors have demonstrated some commitment to experimenting with consistency terms, attempting to tackle an important challenge in generative modeling.

**Weaknesses:**

1.	Insufficient Justification for Consistency Terms: The paper lacks a solid theoretical foundation for enforcing consistency terms at each diffusion step, which may not align with the iterative nature of the diffusion process. This raises concerns about the model’s validity.
	2.	Impact on Variance of Generative Model: Enforcing consistency terms across all steps could reduce the model’s generative diversity, possibly resulting in outputs that lack the variability expected in a robust generative framework. This could push the model towards a U-Net-like structure, potentially sacrificing the inherent variability necessary for effective 3D generation.
	3.	Experimental Limitations: The experiments do not convincingly demonstrate that this approach generalizes well. The benchmarks and comparative studies are limited, and it is unclear if the performance improvements observed are due to the proposed model’s consistency terms or other factors.
4. I believe equation 5 is incorrect. The variational bound term should have the joint probability  p(x_0 : x_t)  as the numerator.

**Questions:**

1.	Could the authors provide a theoretical basis for enforcing consistency terms across all diffusion steps, rather than focusing on key steps where 3D structure becomes clearer?
	2.	How does the addition of consistency terms impact the model’s ability to generate diverse outputs? Is there a measurable loss in generative variability?
	3.	Could the authors elaborate on why 3D priors, rather than other forms of regularization, are necessary for improving consistency?

---

### Official Review · Reviewer_EKmn · 2024-11-04

**Soundness:** 3
**Presentation:** 3
**Contribution:** 2
**Rating:** 5
**Confidence:** 3

**Summary:**

The paper introduces a new way to generate 3D point clouds from a single image. The main contribution of the paper is to use 2D and 3D priors as a way to nudge diffusion models for robustness in the ill-posed problem domain of single view 3D point cloud estimation.
2D priors are extracted from DINOv2, as depth and contour (as well as features). 3D priors are extracted as random camera transformations around an object of interest.
The results demonstrate that the method can outperform existing methods for single view point cloud estimation.

**Strengths:**

Few strength of the papers that the reviewer appreciates are:
1. The paper is fairly well written and easy to follow.
2. The contributions are clearly isolated, from 2D and 3D, making it easy to justify the quality of each contributions in the ablation in the tables, e.g, Table 4
3. Numerous technical ablations conducted to demonstrate how each components and their respective small deviations work.

**Weaknesses:**

Main weakness of the paper is that the contributions are not as well novel. Usage of 2D features and derivatives such as depth and contours are easily justifiable. However, because it is so clear and evident that use of depth and contour as a way to regularize 3D point cloud reconstruction will help, the reviewer does not find it as fundamental contribution to the community.
In other words, yes we know that the usage of depth and contours will help, and yes the paper has re-verified it. What leaves the takeaways? The reviewer is unsure if usage of these priors as a form of augmentations are worthy of contributions to the ICLR venue.

In addition, other contributions, such as usage of consistency in the diffusion process is not new; it may be new in the realm of point cloud diffusion, but it would be application of existing approaches.

**Questions:**

Few questions that the reviewer has for the paper:
1. How does the model perform on non-object centric scenes? (i.e, more extrapolating views)
2. How are random camera parameters sampled? How did the authors make sure that there is no bias brought in on the sampling procedure of the camera parameters?

Comment:
Typo in Figure 2 (a).

---

### Official Review · Reviewer_mAdu · 2024-11-08

**Soundness:** 3
**Presentation:** 3
**Contribution:** 3
**Rating:** 6
**Confidence:** 4

**Summary:**

This paper presents a novel Consistency Diffusion Model (CDM) designed to improve single-image 3D reconstruction. The proposed model leverages both 2D and 3D prior information to enhance the consistency of the reconstruction process, yielding promising results in single-view scenarios. Experimental evaluations demonstrate that CDM outperforms existing methods in certain cases, underscoring its potential.

**Strengths:**

1.This paper is easy to follow and well-written.
2.The paper introduces the Consistency Diffusion Model, which incorporates 3D structural priors as constraints within a diffusion framework. The topic is interesting.
3.The model employs a Bayesian framework, incorporating a new constraint term that introduces 3D structural priors into the variational process. This improves the model's consistency by raising the Evidence Lower Bound (ELBO), reducing uncertainty, and enhancing overall stability.

**Weaknesses:**

1.The inclusion of multiple priors, complex rotation matrices, and depth mapping computations increases the computational burden during training. There is a lack of detailed information on the training time and computational efficiency.
2.Additional experiments incorporating different types of 3D structural priors and 2D image priors, as well as testing on a broader range of datasets, would help to validate the model’s generalizability and robustness across diverse conditions.
3.The paper notes that inconsistent conditions between the training and sampling phases can lead to "model drift," causing learning biases and unstable results. This could result in a performance gap between the training and deployment phases, affecting the model's real-world reliability. However, potential methods for mitigating or addressing the issue of model drift are not discussed.

**Questions:**

Please see the Weaknesses.

---

### Note · Authors · 2024-11-13

I have read and agree with the venue's withdrawal policy on behalf of myself and my co-authors.